# Effects of RIPC on the Metabolome in Patients Undergoing Vascular Surgery: A Randomized Controlled Trial

**DOI:** 10.3390/biom12091312

**Published:** 2022-09-16

**Authors:** Kadri Eerik, Teele Kasepalu, Karl Kuusik, Jaan Eha, Mare Vähi, Kalle Kilk, Mihkel Zilmer, Jaak Kals

**Affiliations:** 1Endothelial Research Centre, University of Tartu, 8 Puusepa Street, 51014 Tartu, Estonia; 2Department of Cardiology, Institute of Clinical Medicine, University of Tartu, 8 Puusepa Street, 51014 Tartu, Estonia; 3Heart Clinic, Tartu University Hospital, 8 Puusepa Street, 51014 Tartu, Estonia; 4Institute of Mathematics and Statistics, University of Tartu, 18 Narva mnt. Street, 51009 Tartu, Estonia; 5Department of Biochemistry, Institute of Biomedicine and Translational Medicine, Centre of Excellence for Genomics and Translational Medicine, University of Tartu, 19 Ravila Street, 50411 Tartu, Estonia; 6Department of Surgery, Institute of Clinical Medicine, University of Tartu, 8 Puusepa Street, 51014 Tartu, Estonia; 7Surgery Clinic, Tartu University Hospital, 8 Puusepa Street, 51014 Tartu, Estonia

**Keywords:** metabolomics, remote ischemic preconditioning, vascular surgery

## Abstract

Background: remote ischemic preconditioning (RIPC) is a phenomenon in which short episodes of ischemia are applied to distant organs to prepare target organs for more prolonged ischemia and to induce protection against ischemia-reperfusion injury. This study aims to evaluate whether preoperatively performed RIPC affects the metabolome and to assess whether metabolomic changes correlate with heart and kidney injury markers after vascular surgery. Methods: a randomized sham-controlled, double-blinded trial was conducted at Tartu University Hospital. Patients undergoing elective open vascular surgery were recruited and RIPC was applied before operation. Blood was collected preoperatively and 24 h postoperatively. The metabolome was analyzed using the AbsoluteIDQ p180 Kit. Results: final analysis included 45 patients from the RIPC group and 47 from the sham group. RIPC did not significantly alter metabolites 24 h postoperatively. There was positive correlation of change in the kynurenine/tryptophan ratio with change in hs-troponin T (*r* = 0.570, *p* < 0.001), NT-proBNP (*r* = 0.552, *p* < 0.001), cystatin C (*r* = 0.534, *p* < 0.001) and beta-2-microglobulin (*r* = 0.504, *p* < 0.001) only in the RIPC group. Conclusions: preoperative RIPC did not significantly affect the metabolome 24 h after vascular surgery. The positive linear correlation of kynurenine/tryptophan ratio with heart and kidney injury markers suggests that the kynurenine–tryptophan pathway can play a role in RIPC-associated cardio- and nephroprotective effects.

## 1. Introduction

Remote ischemic preconditioning (RIPC) is a phenomenon in which short controlled non-lethal episodes of ischemia are applied to distant organs or limbs to prepare target organs for more prolonged ischemia and to induce protection against ischemia-reperfusion injury [1]. Previous studies and meta-analyses have investigated the effect of RIPC in both cardiac and noncardiac surgery, but the beneficial effect of RIPC remains inconclusive [2]. Vascular surgery involves clamping of the main arterial supply to a particular territory; therefore, these patients are potentially prone to ischemic insult and ischemia-reperfusion (IR) injury. Hence, we and other researchers have been interested in RIPC as a possible method to reduce organ damage in vascular surgery patients [2,3,4]. A recent systematic review and meta-analysis showed no significant benefit of RIPC on different clinical parameters [2]. However, the studies included in the meta-analysis were relatively small and heterogeneous; therefore, potential effects may have been missed.

Understanding RIPC’s precise mechanisms of action can help draw solid conclusions about its usefulness in reducing preclinical and clinical injury to ischemia-sensitive organs. Although extensively studied, the exact mechanisms of RIPC remain unclear. It is suggested that humoral and neural mechanisms, circulating immune cells, and activation of hypoxia-inducible genes may facilitate its effects [1]. Less focus has been given to potential alterations in patients’ metabolomes following RIPC [5]. Metabolomics can provide new insights into understanding the mechanisms of RIPC.

It is suggested that the effects of RIPC are mediated by changes in amino acid and energy metabolism. Studies on rats have shown that alanine, aspartate, glutamate, arginine, and proline metabolism might be involved in the beneficial effects of RIPC [6,7,8]. RIPC-exposed rats were able to partially normalize their altered energy metabolism and shift the organism to ketone body synthesis to provide more energy to ischemia-sensitive organs such as the brain and the heart [7]. A ketone body, 3-hydroxybutyrate, has been shown to facilitate the effects of RIPC, since its levels normalize within 24 h of ischemia in RIPC rats. This is in contrast with non-RIPC rats, whose levels of 3-hydroxybutyrate remain low [7].

Along with participating in energy metabolism, some amino acids mediate the effects of RIPC through different pathways. RIPC has been associated with increased carnosine levels [8]. Carnosine has well-known antioxidant properties [9]. Following RIPC, higher levels of kynurenine have been detected [8]. Kynurenine is considered to be a vasoactive metabolite of tryptophan, which can dilate coronary vessels and hence play a role in RIPC-mediated cardioprotection [10]. In addition to amino acids, other small molecules can mediate the effects of RIPC. Several lipid-related metabolites, e.g., palmitic, stearic, oleic, and linoleic acid can stimulate toll-like receptor 4 (TLR4) signaling, activating pro-inflammatory pathways [6,11]. These metabolites were upregulated in IR injury but were downregulated by RIPC, suggesting that RIPC could have anti-inflammatory effects [6].

Several of RIPC’s beneficial effects in organ protection have been found to be mediated through pathways related to amino acid and energy metabolism [5,6,7,8]. However, extensive investigation is still needed, since most of the studies explaining the effects of RIPC on metabolism have been conducted on rats and need to be validated in human settings. This paper aims to describe perturbations in the plasma metabolome following RIPC in patients undergoing major vascular surgery and to assess correlations between shifts in the metabolome and heart and kidney damage markers.

## 2. Materials and Methods

### 2.1. Eligibility and Study Groups

From 1 January 2016, to 8 February 2018, a randomized sham-controlled double-blind clinical trial was conducted at Tartu University Hospital’s Clinic of Surgery, Department of Vascular Surgery [3,4,5].

Patients undergoing elective open surgical repair of an infra-renal abdominal aortic aneurysm (AAA), carotid endarterectomy or surgical lower limb revascularization surgery (common femoral endarterectomy, aorto (bi) femoral or femoropopliteal or femorotibial or iliofemoral bypass surgery) were enrolled in the study. Each patient signed an informed consent form prior to the study. The study’s research protocol was approved by the University of Tartu’s Research Ethics Committee and entered in the ClinicalTrials.gov database (NCT02689414).

Age under 18 years, pregnancy, malignancy in the previous five years, permanent atrial fibrillation or flutter, symptomatic upper limb atherosclerosis, home oxygen therapy, preoperative estimated glomerular filtration rate (eGFR) under 30 mL/min/1.73 m^2^, previous history of upper limb vein thrombosis or vascular surgery in the axillary region, and inability to follow the study regimen were the exclusion criteria.

### 2.2. Randomization

Patients were allocated to the sham or RIPC groups at random and in equal numbers. A stratified block design with a block size of 2 or 4 was used. The computer application WINPEPI (PEPI-for-Windows) generated a random sequence. Patients were stratified based on their age (under or over 65 years) and physical status classes 2, 3, or 4 of the American Society of Anesthesiologists (ASA). Randomization and opaque sealed envelopes were prepared by a third party. The envelopes were opened right before intervention.

### 2.3. Intervention

The RIPC protocol comprised four 5-min ischemia episodes followed by a 5-min reperfusion phase. Ischemia was induced by placing a blood pressure cuff on the patient’s arm and elevating cuff pressure to 200 mmHg. If the patient’s blood pressure was above 180 mmHg, cuff pressure was increased to 20 mmHg higher than the patient’s systolic blood pressure. The achievement of successful ischemia was confirmed by witnessingskin color change and monitoring of the patient’s blood pressure on the other arm to maintain sufficient pressure in the cuff. In the sham group cuff pressure was kept at the same level as venous pressure (10–20 mmHg). Intervention began along with the operating room’s preparation for anesthesia. Other components of surgery, such as anesthesia and medication administration, were unaffected.

### 2.4. Blinding

The study intervention was blinded from the patient, his or her physician, surgeon, anesthesiologist, and the rest of the surgical team. The scale of the manometer was kept covered. The statistician was not aware of the significance of the group affiliation.

### 2.5. Outcomes

Blood samples for the analysis of metabolites were taken on the morning of surgery and approximately 24 h postoperatively. The last blood collection was set as close to 24 h post-surgery as possible, on the condition that the patient had fasted for at least three hours. The blood samples were centrifuged, and serum was separated and stored at −80 °C in the refrigerator.

The AbsoluteIDQp180 kit (Biocrates Life Sciences AG, Innsbruck, Austria) was used to measure the levels of the different metabolites. The analytical procedure was performed according to the manufacturer’s standard protocol at the laboratory of the Department of Biochemistry, University of Tartu. Measurements were carried out using a QTRAP 4500 (ABSciex, Framingham, MA, USA), linked to an Agilent 1260 series HPLC (Agilent Technologies, Santa Clara, CA, USA) with a C18 column, and flow injection analysis.

Blood samples were taken preoperatively and about 24 h after surgery for measurement of high sensitivity troponin T (hs-TnT), N-terminal pro-brain natriuretic peptide (NT-proBNP), creatinine (Cr), cystatin C (cysC), and beta-2-microglobulin (B2M), which were used to calculate correlations. The levels of cardiac and kidney function biomarkers were measured at Tartu University Hospital’s United Laboratories.

Sandwich electrochemiluminescence immunoassays (ECLIA), specifically the Elecsys troponin T high-sensitive assay, STAT version (Roche Diagnostics, Penzberg, Bavaria, Germany), and the Elecsys proBNP II (Roche Diagnostics, Penzberg, Bavaria, Germany) were employed according to the manufacturer’s protocol for analysis of hs-TnT and NT-proBNP. The Creatinine plus ver. 2 enzymatic method (Roche Diagnostics, Penzberg, Bavaria, Germany), the Tina-quant Cystatin C Gen. 2 particle enhanced immunoturbidimetric assay (Roche Diagnostics, Penzberg, Bavaria, Germany), and the Tina-quant β2-Microglobulin (serum/plasma application) immunoturbidimetric assay method (Roche Diagnostics, Penzberg, Bavaria, Germany) were used according to the manufacturer’s instructions for assessment of creatinine, cystatin C, and beta-2-microglobulin, respectively.

All patients were enquired about their medical history and medications. An electronic health database and surgery protocols were used to obtain a thorough anamnesis.

### 2.6. Statistical Analysis

The RIPC and sham groups were compared using Student’s *t*-test, and the Wilcoxon rank-sum or Chi-square test. Student’s *t*-test was used in the case of a normal distribution and the Wilcoxon rank-sum test was applied in the case of a non-normal distribution. The Kolmogorov-Smirnov test was used to test for normality. Correlations were calculated using the Pearson or Spearman correlation coefficient. *p*-values below 0.05 were considered significant for comparison of baseline characteristics. Because of multiple comparisons, the Benjamini-Hochberg procedure was used to control false discovery rate. According to the Benjamini-Hochberg procedure, a new significance level was calculated, and *p*-values below 0.0012 were considered significant for comparing changes in the metabolites and assessing correlations of the metabolites with heart and kidney injury markers. Metabolite enrichment analysis was performed with MetaboAnalyst 5.0 by using SMPDB and KEGG databases for which at least two intermediates were present.

## 3. Results

### 3.1. Overview of the Study Groups

A total of 98 patients were enrolled and randomized into study groups. Final analysis included 45 patients from the RIPC group and 47 patients from the sham group. Detailed patients’ flow is shown in Figure 1.

The median time from the end of the intervention to the beginning of surgery did not differ significantly (*p* = 0.057) between the RIPC (36 min, IQR 21–46 min) and the sham group (25 min, IQR 15–38 min). The baseline characteristics of patients did not differ significantly between the study groups (Table 1).

### 3.2. Changes in the Metabolites 24 h Postoperatively

The AbsoluteIDQ^®^ p180 Kit allows us to identify 188 different metabolites: 21 amino acids, 21 biogenic amines, one carbohydrate molecule, 40 (acyl-) carnitines, 14 lysophosphatidylcolines, 76 phosphatidylcholines, and 15 sphingomyelins. In this study, 103 of these metabolites and 20 metabolic ratios were included in the final analysis. A complete list of the analyzed metabolites is provided in Table 2. All (acyl-) carnitines were excluded as they had been discussed in a separate paper by the same authors [5]. Forty-five more metabolites were excluded, as >33% of their pre- or postoperative values were below the level of detection (<LOD).

The baseline values of the analyzed metabolites were similar in both groups (Table 2 and Table 3). The exact values for all metabolites are provided in Appendix A. The aim of this study was to evaluate whether RIPC affects the patients’ metabolic profile 24 h after the surgery. However, there were no statistically significant differences between the groups in changes in the metabolites 24 h postoperatively (Table 2, Table 3 and Appendix A). In quantitative enrichment analysis, none of the metabolic pathways in SMPDB or KEGG databases were found to differ significantly in their response due to RIPC.

### 3.3. Correlations of the Metabolites with Cardiac and Kidney Markers in the RIPC Group

In the RIPC group, there were no significant correlations between change in amino acid levels and change in heart injury markers. Regarding biogenic amines, change in total DMA level was positively correlated with shift in hs-TnT (*r* = 0.496, *p* < 0.001). Changes in different glycerophospholipids and sphingolipids did not correlate with changes in cardiac markers. Among the metabolic ratios, change in the kynurenine/tryptophan ratio showed positive linear correlation with change in hs-TnT (*r* = 0.570, *p* < 0.001) and change in NT-proBNP (*r* = 0.552, *p* < 0.001). These correlations were only seen in the RIPC group.

In the RIPC group, there were no significant correlations of change in amino acids, biogenic amines, glycerophospholipids, or sphingolipids with change in kidney injury markers. However, among the metabolic ratios, there was a significant positive linear correlation of change in the kynurenine/tryptophan ratio with cystatin C (*r* = 0.534, *p* < 0.001) and beta-2-microglobulin (*r* = 0.504, *p* < 0.001). These correlations were not observed in the sham group.

### 3.4. Correlations of the Metabolites with Heart and Kidney Markers in the Sham Group

In the sham group, no significant correlations were detected between alterations in amino acids and heart and kidney injury markers. Nor were there significant correlations between change in biogenic amines and change in hs-TnT or NT-proBNP. However, there was a significant positive linear correlation between change in total DMA and the change in creatinine (*r* = 0.462, *p* = 0.001) and beta-2-microglobulin (*r* = 0.491, *p* < 0.001).

There occurred no significant correlation of different glycerophospholipid and sphingolipid level with change in heart and kidney injury markers. Among the metabolic ratios, there was a significant positive linear correlation between change in the ADMA/Arg ratio and change in hs-TnT (*r* = 0.527, *p* < 0.001), between change in the Cit/Arg ratio and change in beta-2-microglobulin (*r* = 0.500, *p* < 0.001) and, between change in the putrescine/ornithine ratio and change in creatinine (*r* = 0.615, *p* < 0.001).

## 4. Discussion

There are few studies evaluating the effect of RIPC on the metabolome in vascular surgery patients. We assessed the metabolic profile and correlations of change in metabolites with heart and kidney markers 24 h following surgery. RIPC did not significantly alter metabolites postoperatively. However, we found a significant positive linear correlation of change in the kynurenine/tryptophan ratio with change in hs-TnT, NT-proBNP, cystatin C, and beta-2-microglobulin in the RIPC group. We have demonstrated that RIPC reduces the leakage of cardiac and kidney injury markers in the same patient cohort and may offer cardio- and nephroprotection [3,4]. The tryptophan-kynurenine (TRP-KYN) pathway has been associated with RIPC in several studies, and it is suggested that RIPC decreases tryptophan levels and increases kynurenine levels in the blood, and that kynurenine injection is cardioprotective [8,12].

Under normal physiological conditions, the metabolism of tryptophan follows two pathways: the serotonin (5-hydroxytryptamine, 5-HT) and the kynurenine pathway [13,14]. Only a small amount of tryptophan is used for 5-HT biosynthesis in the gut and brain. The majority of tryptophan (up to 95%) enters the TRP-KYN pathway in peripheral organs [13,14]. Several mechanisms have been proposed regarding how RIPC affects the TRP-KYN pathway. The first critical reaction in the TRP-KYN pathway is oxidation of tryptophan into formylkynurenine. In healthy individuals, this reaction is strictly controlled by tryptohan 2,3-dioxygenase (TDO) in the liver and indoleamine 2,3-dioxygenase (IDO) in other tissues [12,13,14]. Under normal conditions, TDO metabolizes 95% of tryptophan within the liver, being the main determinant of distribution of hepatic tryptophan and kynurenine to other tissues. In the liver, kynurenine is subsequently converted into kynurenic acid and other products, which contribute to the restoration of energy supplies via the glutarate and nicotinamide adenine dinucleotide (NAD) pathways [12,13,14]. Following RIPC, the plasma concentration of tryptophan decreases, whereas its concentration in the liver increases [12]. One potential mechanism of this could be increased free tryptophan uptake by the liver following TDO activation. It is argued that RIPC may be related to increased TDO activity by allosteric regulation that does not require protein synthesis [12]. As the liver is the most relevant determinant of kynurenine distribution to other tissues, TDO activation is likely to be compatible with the plasmatic kynurenine increase after RIPC [12].

In addition to the presence of TDO, increased activity of IDO enhances formation of kynurenine and related products [12,15]. IDO expression is principally upregulated by interferon-γ (IFN-γ), and activation of the TRP-KYN pathway in response to inflammatory stimuli can be detected based on an increased kynurenine/tryptophan ratio [12,15]. It has been demonstrated that both tryptophan and kynurenine dilate preconstricted porcine coronary arteries in a dose-dependent manner [8,10]. The vasodilatation induced by tryptophan was found to require the contribution of IDO and an intact endothelium, while kynurenine was able to dilate coronary vessels independently of the endothelium [8,15].

These findings suggest that kynurenine is the tryptophan’s cardioprotective metabolite, which could play a role in RIPC-induced cardioprotective effects [8,10,12]. Although our study is not sufficient to explain the exact physiological mechanism underlying RIPC-induced cardioprotection, it can still provide additional evidence about the involvement of the TRP-KYN pathway in it through demonstrating a strong positive linear correlation of the kynurenine/tryptophan ratio with hs-TnT and NT-proBNP.

To our knowledge, there are not many studies evaluating the associations between the TRP-KYN pathway, RIPC, and nephroprotection. However, a recent study on mice found that hypoxic preconditioning increases serum kynurenine levels and stimulates kynurenine biotransformation, leading to the preservation of NAD^+^ in the post-ischemic kidney [16]. NAD^+^ via cellular energy restoration has a critical role in renal resistance against ischemic insults by connecting oxidative metabolism in the epithelium to overall organ function [16].

The TRP-KYN pathway also plays a role in acute kidney injury (AKI). The TRP-KYN pathway is activated in AKI, showing usually an increase in its metabolites [17,18]. Upregulation of the TRP-KYN pathway in AKI can be explained by the higher concentrations of IFN-γ and other pro-inflammatory cytokines following IR injury, which in turn activate the TRP-KYN pathway via its rate-limiting enzyme IDO [19]. In our study a strong linear correlation of change in the kynurenine/tryptophan ratio with change in cystatin C and beta-2-microglobulin was seen in the RIPC group. Further studies are needed to investigate exact pathophysiological associations between acute kidney injury, RIPC’s potential nephroprotective effects, and the TRP-KYN pathway. Other RIPC’s nephroprotective effects have been shown to be mediated via amino acid, especially L-alanine, and some lipid-related metabolites metabolism [6]. Still, our study failed to find these correlations.

Another metabolite that had several correlations with different heart and kidney markers in both groups was total dimethylarginine (DMA). Total DMA is the sum of asymmetric DMA (ADMA) and symmetric DMA (SDMA) [20]. ADMA is an endogenous nitric oxide synthase (NOS) inhibitor and hence a known mediator of endothelial cell dysfunction, oxidative stress, and atherosclerosis [21]. Circulating levels of ADMA have been found to correlate with well-known cardiovascular risk factors and higher risk of cerebrovascular events such as acute ischemic stroke [21]. In the RIPC group, total DMA was positively linearly correlated with hs-TnT and in the sham group, with creatinine and beta-2-microglobulin. Also, in the sham group, there was a significant positive correlation between a change in the ADMA/Arg ratio and change in hs-TnT. This suggests that total DMA can play a role in ischemia-reperfusion and surgery-related heart and kidney injury. It has potential as an early biomarker in predicting heart and kidney damage. Its function and association in terms of RIPC require further investigation. Based on our findings and the literature, possible metabolomic interactions between RIPC and vascular surgery are summarized in Figure 2.

To conclude, this study brought out the above-described correlations of metabolites with heart and kidney injury markers. However, we did not find any statistically significant changes in the metabolites in response to RIPC 24 h postoperatively, which could be explained as follows. First, the patient’s metabolic profile can be affected by different types of surgery that involve different degrees of tissue damage. It can be assumed that patients undergoing open surgical repair of abdominal aortic aneurysm had more serious tissue damage and had to deal with a greater extent of ischemia and reperfusion compared to patients with peripheral artery disease (PAD) undergoing lower-limb revascularization surgery. Therefore, the involvement of different surgeries could have significantly affected the results. However, we tried to minimize it by matching the number of various surgeries across groups. Second, it has also been suggested that patients with different stages of PAD have distinct metabolic profiles [22]. The heterogeneity of different surgeries, as well as the different stages of the disease, could have affected the results. Third, it has been suggested that the common anesthetic propofol has deleterious effects on RIPC [23]. In this study, 42% of the patients in the RIPC group had propofol-induced anesthesia, which could have diminished the effect of RIPC. Fourth, diabetes has been shown to reduce the impact of RIPC [24,25]. However, as there were only 11% of the diabetic patients in the RIPC group, the overall effect of diabetes could not have been pronounced. Fifth, it is possible, too, that due to the heterogeneous study population and the relatively small sample size (92 patients), the subtle effects of RIPC remained unnoticed. It would be fruitful to carry out additional studies on larger sample sizes. Sixth, all patients underwent major surgery, which affected their metabolome. It can be argued that the effects of RIPC could have been shadowed by more substantial metabolomic alterations induced by surgery and tissue trauma. Seventh, it remains controversial as to which RIPC protocol is the best. In this study, a single RIPC procedure consisting of four 5 min cycles of ischemia followed by 5 min of reperfusion was applied. Still, some researchers have recently focused on repeated episodes of RIPC (7 days up to one month), which may prove more effective. Therefore, future studies should assess whether chronic (7 up to 28 days) RIPC will affect the metabolome and could be useful in reducing vascular surgery-related heart and kidney injuries. Eighth, the blood samples were collected preoperatively and 24 h after surgery. It is not yet clear what the most optimal time point is for assessing RIPC-induced metabolomic changes. The mechanism of RIPC is biphasic, involving early and delayed protection. An early type of protection (first “window”) is brief and transient (lasting a few hours after conditioning), whereas delayed protection (second “window”) can last hours to days [1,26]. Therefore, the optimal timing to assess the RIPC effect remains unclear. It can be argued that 24 h later, the effects of RIPC’s first “window” were no longer detectable in the plasma; however, we were able to assess the effects of RIPC’s second “window”. Assumably we would have seen more significant effects of RIPC when we had performed RIPC before the surgery as well as 24 h earlier to target both RIPC’s early and delayed type of protection. Also, all metabolites were measured in the plasma as opposed to the local area of surgery. It is presumable that simultaneous assessment of local and systemic metabolic alterations could better explain metabolic pathways involved in RIPC. Consequently, this study failed to find any statistically significant acute changes following RIPC 24 h postoperatively.

## 5. Conclusions

In this study, preoperatively performed RIPC did not significantly affect patients’ metabolome 24 h after vascular surgery. However, the kynurenine/tryptophan ratio showed significant correlation with heart and kidney injury markers in the RIPC group. These findings suggest that the kynurenine–tryptophan pathway can play a role in RIPC-associated cardio- and nephroprotective effects. Further larger studies would be beneficial for drawing definite conclusions about whether RIPC influences the metabolome and how these metabolomic changes correlate with heart and kidney injury markers.

## Figures and Tables

**Figure 1 biomolecules-12-01312-f001:**
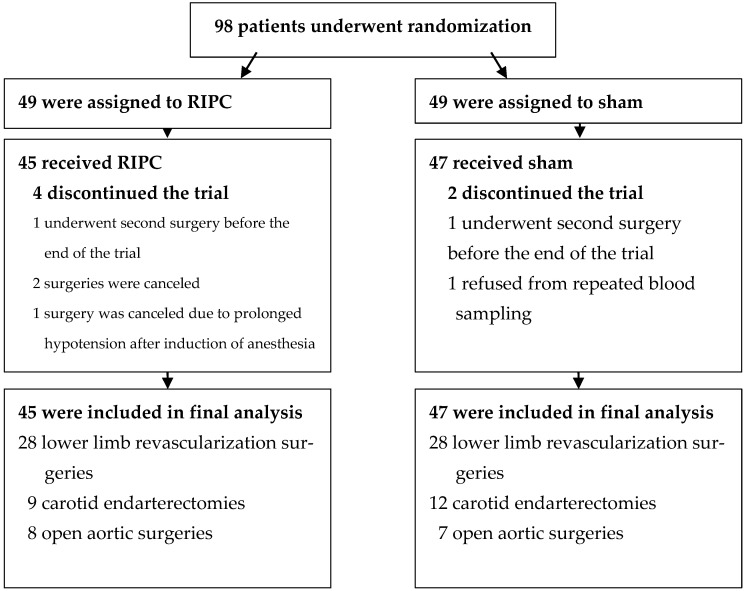
Patients’ flow chart.

**Figure 2 biomolecules-12-01312-f002:**
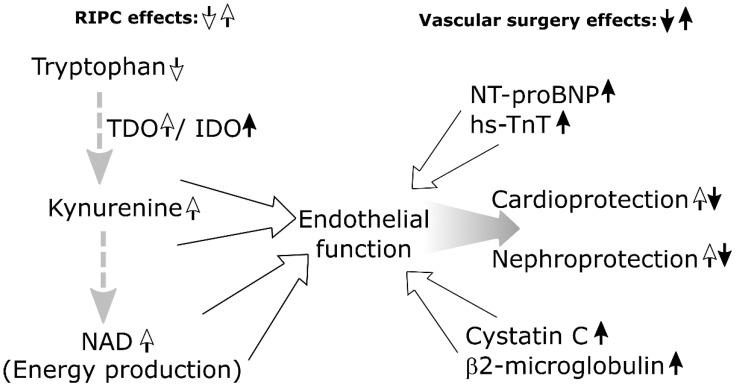
Hypothetical metabolomic interactions of RIPC and vascular surgery based on literature and current report.

**Table 1 biomolecules-12-01312-t001:** Baseline characteristics.

Variable	RIPC (*n* = 45)	Sham (*n* = 47)	*p*-Value
Age, years (SD)	67 (± 9)	66 (± 10)	0.577
Male, *n* (%)	36 (80)	32 (68)	0.288
BMI, kg/m^2^ (SD)	26.3 (± 6.4)	26.5 (± 6.7)	0.840
ASA 2, *n* (%)	18 (40)	19 (40)	1
ASA 3, *n* (%)	20 (44)	22 (47)	0.986
ASA 4, *n* (%)	7 (16)	6 (13)	0.933
ACEI or ARB, *n* (%)	21 (47)	30 (64)	0.148
Calcium channel blockers, *n* (%)	9 (20)	17 (37)	0.135
Beta-blockers, *n* (%)	11 (24)	19 (40)	0.158
Statins, *n* (%)	13 (29)	14 (30)	1
Diabetes, *n* (%)	5 (11)	8 (17)	0.607
Myocardial infarction, *n* (%)	8 (18)	3 (6)	0.172
Stroke, *n* (%)	10 (22)	12 (26)	0.899
Smoker (current or ex-smoker), *n* (%)	40 (89)	42 (89)	1
MAP, mmHg (SD)	99 (± 12)	100 (± 11)	0.678
Heart rate, bpm (SD)	66 (± 9)	67 (± 11)	0.754
Cholesterol, mmol/L (IQR)	5.0 (4.2–5.7)	5.0 (3.9–5.6)	0.793
LDL, mmol/L (IQR)	3.4 (8.1–10.4)	3.3 (2.5–3.8)	0.500
HDL, mmol/L (IQR)	1.1 (0.9–1.4)	1.1 (1.0–1.3)	0.311
Triglycerides, mmol/L (IQR)	1.6 (1.3–1.8)	1.5 (1.2–2.0)	0.787
Administration on propofol, *n* (%)	19 (42)	26 (55)	0.295
Duration of surgery, min (IQR)	108 (89–135)	112 (84–156)	0.827

BMI—body mass index, ASA—American Society of Anesthesiologists’ physical status score, ACEI—angiotensin-converting enzyme inhibitor, ARB—angiotensin II receptor blocker, MAP—mean arterial blood pressure, LDL—low-density lipoprotein, HDL—high-density lipoprotein, SD—standard deviation, IQR—interquartile range. *p*-values were calculated for data with a normal distribution (presented as mean and SD) using Student’s *t*-test; for data with a non-normal distribution (presented as median and IQR) using the Wilcoxon rank-sum test; and for binary data (presented as number and percentage), using the Chi-square test.

**Table 2 biomolecules-12-01312-t002:** Complete list of metabolites included in final analysis and summarized comparison of their baseline and postoperative values between the RIPC and sham groups.

Metabolic Group and Metabolites	Baseline Comparison *	Change 24 h Postoperatively *
Amino acids (*n* = 19) Ala, Arg, Cit, Gln, Glu, Gly, His, Ile, Leu, Lys, Met, Orn, Phe, Pro, Ser, Thr, Trp, Tyr, Val	*p* ˃ 0.001	*p* ˃ 0.001
Biogenic amines (*n* = 7) ADMA, Creatinine, Kynurenine, Serotonine, Spermine, Taurine, total DMA	*p* ˃ 0.001	*p* ˃ 0.001
Glycerophospholipids (*n* = 62) lysoPCaC16:0, lysoPCaC16:1, lysoPCaC17:0, lysoPCaC18:0, lysoPCaC18:1, lysoPCaC18:2, lysoPCaC20:3, lysoPCaC20:4, lysoPCaC26:1, PCaaC28:1, PCaaC30:0, PCaaC32:0, PCaaC32:1, PCaaC32:2, PCaaC32:3, PCaaC34:1, PCaaC34:2, PCaaC34:4, PCaaC36:0, PCaaC36:1, PCaaC36:2, PCaaC36:3, PCaaC36:4, PCaaC36:5, PCaaC38:0, PCaaC38:3, PCaaC38:4, PCaaC38:5, PCaaC38:6, PCaaC40:4, PCaaC40:5, PCaaC40:6, PCaaC42:4, PCaaC42:5, PCaaC42:6, PCaeC30:1, PCaeC32:1, PCaeC32:2, PCaeC34:0, PCaeC34:1, PCaeC34:2, PCaeC34:3, PCaeC36:0, PCaeC36:1, PCaeC36:2, PCaeC36:3, PCaeC36:4, PCaeC36:5, PCaeC38:0, PCaeC38:3, PCaeC38:4, PCaeC38:5, PCaeC38:6, PCaeC40:1, PCaeC40:2, PCaeC40:4, PCaeC40:5, PCaeC40:6, PCaeC42:4, PCaeC44:4, PCaeC44:5, PCaeC44:6	*p* ˃ 0.001	*p* ˃ 0.001
Sphingolipids (*n* = 14) SM(OH)C14:1, SM(OH)C16:1, SM(OH)C22:1, SM(OH)C22:2, SM(OH)C24:1, SMC16:0, SMC16:1, SMC18:0, SMC18:1, SMC20:2, SMC24:0, SMC24:1, SMC26:0, SMC26:1	*p* ˃ 0.001	*p* ˃ 0.001
Hexoses (*n* = 1) H1	*p* ˃ 0.001	*p* ˃ 0.001
Metabolic ratios (*n* = 20) (C2 + C3)/C0, AAA, ADMA/Arg, BCAA, C2/C0, Cit/Arg, Cit/Orn, Essential AA, Fisher ratio, Glucogenic AA, Kynurenine/Trp, Nonessential AA, Orn/Arg, Putrescine/Orn, Serotonin/Trp, Total SM, Total SM-nonOH, Total SM-OH, Total SM-OH/Total SM-nonOH, Tyr/Phe	*p* ˃ 0.001	*p* ˃ 0.001

Ala—Alanine, Arg—Arginine, Cit—Citrulline, Gln—Glutamine, Glu—Glutamic acid, Gly—Glycine, His—Histidine, Ile—Isoleucine, Leu—Leucine, Lys—Lysine, Met—Methionine, Orn—Ornithine, Phe—Phenylalanine, Pro—Proline, Ser—Serine, Thr—Threonine, Trp—Tryptophan, Tyr—Tyrosine, Val—Valine, ADMA—Asymmetric dimethylarginine, DMA—dimethylarginine, lysoPCa—lysoPhosphatidylcholine acyl, PCaa—Phosphatidylcholine diacyl, PCae—Phosphatidylcholine acyl-alkyl, SM (OH)—Hydroxysphingomyeline, SM—Sphingomyeline, H1—hexose, C2—Acetylcarnitine, C3—Propionylcarnitine, C0—Carnitine, AAA—Aminoadipic acid, ADMA—asymmetric dimethylarginine, BCAA—Branched chain amino acids, AA—Amino acids, OH—hydroxy, nonOH—nonhydroxy. * Provided *p*-value applies to each metabolite listed in the metabolic group. Specific *p*-values for each metabolite regarding the comparison of baseline values and changes 24 h postoperatively are provided in Table 3, Appendix A. Benjamini-Hochberg procedure was used to control the false discovery rate, and *p*-values < 0.0012 were considered significant.

**Table 3 biomolecules-12-01312-t003:** Comparison of baseline levels and changes in the metabolites 24 h after operation between the RIPC and sham groups.

	Baseline			Change 24 h Postoperatively		
	Sham	RIPC		Sham	RIPC	
Metabolite	Mean (±SD)/Median (IQR)	Mean (±SD)/Median (IQR)	*p*-Value	Mean (±SD)/Median (IQR)	Mean (±SD)/Median (IQR)	*p*-Value
Ala	392.5 (±91.4)	385.5 (±89.9)	0.715	−25.3 (±120.7)	−11.6 (±123.7)	0.592
Arg	114.3 (±29.6)	120.4 (±35.4)	0.365	−20.0 (±38.7)	−21.0 (−35.8–(−2.0))	0.591
Cit	35.9 (±8.2)	34.2 (±9.8)	0.384	-8.8 (±8.9)	−7.6 (±7.8)	0.496
Gln	812.0 (±132.5)	836.5 (±186.1)	0.470	−150.5 (±170.0)	−161.5 (±189.4)	0.770
Glu	72.9 (55.4–95.9)	57.1 (46.4–74.8)	0.036	−0.7 (−15.5–14.6)	−12.7 (−24.0–14.0)	0.128
Gly	240.0 (189.0–286.0)	247.0 (202.0–288.0)	0.885	−26.3 (±53.4)	−22.4 (±52.7)	0.723
His	91.8 (±19.4)	94.7 (±18.7)	0.479	−12.7 (±13.4)	−14.7 (±18.6)	0.550
Ile	85.0 (72.8–103.0)	85.5 (70.6–108.0)	0.867	−16.4 (±28.9)	−25.0 (±32.7)	0.188
Leu	180.0 (147.0–203.0)	168.0 (144.0–205.0)	0.680	−28.2 (±54.6)	−46.1 (±52.3)	0.112
Lys	251.9 (±62.3)	271.0 (±71.0)	0.171	−50.3 (±65.5)	−56.5 (±63.2)	0.645
Met	23.0 (±5.6)	24.5 (±6.4)	0.232	−1.6 (±7.3)	−1.2 (±9.6)	0.822
Orn	96.3 (±26.0)	97.4 (±23.5)	0.826	−25.2 (±30.6)	−26.0 (±24.8)	0.894
Phe	70.5 (63.7–83.3)	72.1 (65.8–78.3)	0.697	2.2 (±12.7)	−1.1 (±14.8)	0.255
Pro	204.7 (±49.7)	206.6 (±63.3)	0.877	−13.0 (±55.0)	−17.0 (−42.0–21.0)	0.666
Ser	134.0 (±34.9)	133.1 (±31.4)	0.893	−27.1 (±37.9)	−32.0 (−53.8–(−10.0))	0.222
Thr	147.0 (109.0–251.0)	111.0 (92.5–161.0)	0.072	−21.6 (±49.1)	−20.9 (−81.0–12.0)	0.516
Trp	64.9 (54.3–76.4)	66.0 (53.9–74.4)	0.885	−8.8 (±17.4)	−10.0 (±15.2)	0.725
Tyr	64.5 (±12.7)	70.9 (±15.1)	0.028	−3.9 (±16.1)	−5.3 (±18.9)	0.692
Val	266.5 (±53.0)	265.5 (±71.7)	0.941	−22.2 (±74.6)	−37.8 (±72.9)	0.315
ADMA	0.6 (±0.2)	0.6 (±0.1)	0.741	−0.1 (±0.2)	−0.1 (±0.2)	0.833
Creatinine	105.0 (81.3–151.0)	108.0 (76.4–143.0)	0.640	1.0 (−9.0–20.0)	-3.7 (−15.0–22.0)	0.322
Kynurenine	0.04 (0.03–0.05)	0.04 (0.03–0.05)	0.353	0.1 (±0.7)	0.1 (±1.0)	0.842
Serotonine	0.4 (0.3–0.6)	0.5 (0.03–0.7)	0.421	−0.1 (±0.1)	−0.1 (±0.1)	0.717
Spermine	0.0 (0.0–3.8)	0.0 (0.0–3.8)	0.711	0.00 (0.00–0.00)	0.00 (0.00–0.00)	0.030
Taurine	107.0 (±30.0)	105.6 (±32.0)	0.827	−14.9 (±30.1)	−10.3 (±31.4)	0.475
Total DMA	1.1 (±0.3)	1.1 (±0.3)	0.330	−0.08 (±0.3)	−0.09 (±0.4)	0.932
H1	4825.0 (4245.0–5231.0)	4505.0 (4207.0–5013.0)	0.128	1231.6 (±1798.1)	1157.7 (±1431.5)	0.828
AAA	204.2 (±35.7)	211.7 (±36.8)	0.325	−10.4 (±37.0)	−16.3 (±38.5)	0.453
ADMA/ Arg	0.01 (0.00–0.01)	0.01 (0.00–0.01)	0.841	0.00 (0.00–0.00)	0.00 (0.00–0.00)	0.776
BCAA	532.2 (±96.7)	537.9 (±143.3)	0.825	−104.0 (−183.0–(−29.0))	−81.0 (−159.0–53.0)	0.238
Cit/Arg	0.3 (±0.1)	0.3 (±0.1)	0.276	−0.02 (±0.11)	0.00 (±0.11)	0.422
Cit/Orn	0.4 (±0.1)	0.4 (±0.1)	0.342	0.04 (−0.07–0.09)	0.00 (−0.12–0.14)	0.474
Essential AA	1087.5 (±208.4)	1081.5 (±212.4)	0.893	−160.0 (±254.7)	−185.4 (±241.4)	0.625
Fisher ratio	2.6 (±0.4)	2.5 (±0.5)	0.326	−0.3 (±0.6)	−0.3 (±0.6)	0.456
Glucogenic AA	740.0 (690.0–872.0)	765.0 (701.0–839.0)	0.770	−84.8 (±162.9)	−57.8 (±175.6)	0.445
Kynurenine/Trp	0.04 (±0.02)	0.04 (±0.01)	0.654	0.01 (0.00–0.01)	0.01 (0.00–0.02)	0.741
Nonessential AA	2319.6 (±252.8)	2346.0 (±338.1)	0.671	−339.9 (±362.3)	−324.0 (±413.2)	0.845
Orn/Arg	0.8 (0.7–1.1)	0.8 (0.6–1.0)	0.741	−0.1 (−0.2–0.1)	−0.1 (−0.3–0.1)	0.901
Serotonin/Trp	0.01 (0.01–0.01)	0.01 (0.01–0.01)	0.391	0.00 (0.00–0.00)	0.00 (0.00–0.00)	0.991
Tyr/Phe	0.9 (±0.2)	1.0 (±0.2)	0.115	−0.1 (±0.2)	−0.1 (±0.2)	0.692

All metabolites are measured in µmol/L, except for metabolic ratios, which do not have a unit. In the case of a normal distribution (Kolmogorov-Smirnov’s test), mean and standard deviation (SD) are given. In the case of a non-normal distribution, median and quartiles (Q1, Q3) are provided. Benjamini-Hochberg procedure was used to control the false discovery rate, and *p*-values < 0.0012 were considered significant. SD—standard deviation, IQR—interquartile range, Ala—Alanine, Arg—Arginine, Cit—Citrulline, Gln—Glutamine, Glu—Glutamic acid, Gly—Glycine, His—Histidine, Ile—Isoleucine, Leu—Leucine, Lys—Lysine, Met—Methionine, Orn—Ornithine, Phe—Phenylalanine, Pro—Proline, Ser—Serine, Thr—Threonine, Trp—Tryptophan, Tyr—Tyrosine, Val—Valine, ADMA—Asymmetric dimethylarginine, DMA—dimethylarginine, H1—hexose, AAA—Aminoadipic acid, ADMA—asymmetric dimethylarg-inine, BCAA—Branched chain amino acids, AA—Amino acids.

## Data Availability

The data sets used in the study are available on request from the corresponding author.

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
