# Peer review of "Effects of RIPC on the Metabolome in Patients Undergoing Vascular Surgery: A Randomized Controlled Trial"

_biomolecules, 2022, doi:10.3390/biom12091312_

Round 1
Reviewer 1 Report
In their paper entitled "Effects of RIPC on the metabolome in patients undergoing vascular surgery: a randomized controlled trial," the authors Kadri E. et al. present an intriguing study examining the potential impact and function of remote ischemic preconditioning (RIPC) to help protect target organs against prolonged and more severe ischemic events as well as ischemia-reperfusion injury. A well-planned experimental strategy supports the author's hypothesis and findings, and the bibliography is consistent with the field.
However, there are some issues with it.
The reviewer is interested in knowing how many replicates were used in each experiment.
It is also necessary to validate the metabolomic results by using a different technique on genes or proteins involved in kynurenine/tryptophan metabolism
What about the outcomes of the Metabolite Set Enrichment Analysis? If feasible, add the results for the implicated route (Metaboanalyst).
To better comprehend the kynurenine/tryptophan metabolism and hypothesis, a graph illustrating the possible pathways would be useful.
Author Response
Dear Reviewer 1,
Thank you for the review and comments. Please see the attachment for our answers.
Kind regards,
Kadri Eerik

Reviewer 2 Report
The aim of this manuscript is to investigate the effects of remote ischemic preconditioning (RIPC) in metabolome and several cardiac and renal makers expression before and after open vascular surgery. The author found that no significant metabolic changes after the RIPC, but some correlation between kynurenine/tryptophan ration with some cardiac and renal makers.
1. How to confirm the successful of Ischemia by using BP cuff? Is there any skin color change and any arterial BP monitor change during the ischemia? Otherwise, markers that proved the successful ischemia should be provided.
2. Blood sample after the RIPC and before the procedure should be collected and used as comparison.
3. Cardiac and renal function markers should be presented in Table. Does RIPC had any effect of those markers, any protective effect?
4. In Table 2, what does the “p>0.001” means? What is the Confidence interval? Please clarify.
5. Comparison between RIPC and Sham group should be done. Any significant protective effect of RIPC between these two groups?
Author Response
Dear Reviewer 2,
Thank you for the review and comments. Please see the attachment for our answers.
Kind regards,
Kadri Eerik

Reviewer 3 Report
In this manuscript, the authors have measured plasma metabolites and injury markers from sham and RIPC vascular surgery patients. Understanding how RIPC compares and contrasts to local ischemic preconditioning is a question with important clinical implications. Although limited effects (differences) are found, the authors are commended for reporting negative data. Nonetheless, limited evaluation of relationships among patient types, poor data presentation, and a limited discussion limit the impact of this paper. Specific comments are detailed below:
Major
1. Although the authors did a good job matching the number of various surgeries across groups (sham vs. RIPC), all surgical procedures were included in one analysis. Despite the critical factor of examining systemic effects of RIPC, could the type of surgery effect the metabolome as well (particularly at 24h)? Although alluded to as a limitation, could the authors address this concern.
2. In this study, ischemia was induced by an increase in blood pressure on the patient’s arm. However, no indication of the change in blood flow to the tissue is measured to evaluate the effectiveness of the ischemia protocol. As limited effects of RIPC are found, could a minimal ischemia be the cause of this?
3. The authors apply their ischemic preconditioning to the arm with the surgical ischemia/reperfusion located at aorta/carotid, renal, or lower leg vessels, with RIPC being the aim of the study. However, with the aim of the study evaluating systemic effects on a distal surgury, could time following RIPC be a factor limiting detection of differences between groups (i.e. the potential protective factors from RIPC have less time to cannot reach surgery site to improve outcome). Further, discussion of how remote IPC differs from local IPC should be included to provide greater context within the discussion.
4. In the results (sections 3.3 and 3.4) correlations are made to cardiac and renal injury markers, however, the data for these markers is not included within the manuscript.
5. All metabolite and injury markers are measured in the blood as opposed to the local area of surgery. Further discussion of how systemic changes from RIPC could alter local metabolism (at surgery site) independent of changes in blood metabolites may benefit the study presentation.
Minor
1. In Table 2, the authors use p>0.001 to indicate a lack of statistical significance. It would be beneficial to provide specific p values as done in all the other tables in the manuscript.
2. While it is understandable that a supplemental table (S1 and S2) is necessary for all the specific comparisons, it makes it seem unnecessary to include the list of all the specific biomolecules in Table 2 (within the main manuscript) as the lists of glycerophospholipids and sphingolipids are particularly long.
3. Section “3.5” should be “3.4”.
Author Response
Dear Reviewer 3,
Thank you for the review and comments. Please see the attachment for our answers.
Kind regards,
Kadri Eerik

Round 2
Reviewer 1 Report
In the reviewer’s opinion, the amount of work after this second round of review is not sufficient to entirely support the conclusions. It appears that the work is just confirmed by the statistics and not verified by the experimental metabolomic data.
In fact, the paper is not supported by biological relevant results, which are also confirmed by the MSEA non-significant outcome. The reviewer is not saying that the work is not interesting; he is only saying that due to its speculative nature it needs more research and results. Moreover the researcher are vague about the question about the techinical replicates. The reviewer understands that the number of biological replicates is one in this kind of experimental plan however, the technical replicates number must be at least three, which has been not specified in this updated version of the paper.
Author Response
We agree that our conclusions should not be taken as an undisputable truth, and additional experimental evidence needs to be collected to completely understand all potential molecular interactions between RIPC, surgery and metabolome. However, we do present novel experimental findings of clinical research from randomized controlled trial (best format of clinical study), combine them, in the best way we can, with existing knowledge and draw conclusions from that. Due to the clinical nature and design of our study, we have also underlined in the manuscript the necessity of future research for a definite understanding of RIPC molecular mechanisms.
Regarding replicate numbers it is actually the technical replicates (e.g. blood sampling, surgery) which cannot be performed on the same subject more than once and are thus equal to one. Biological replicates are defined by the number of distinct biological subjects (patients) who receive the treatment. To best our knowledge, the biological replicates are thus 45, as written in study group description.
Reviewer 2 Report
The author response to all the reviewer's comments with modification. This current version should be consider to publish in the journal.
Author Response
We thank the reviewer for these kind words.
Reviewer 3 Report
The present study by Eerik and colleagues seeks to understand how remote ischemic preconditioning (RIPC) effects the metabolome in blood samples from patients that underwent vascular surgery. Although improvements have been made to the revised manuscript, significant concerns remain regarding experimental methodology and discussion and interpretation of results. These missing components make it challenging to verify the conclusions arrived at by the authors.
Author Response
The present study’s primary endpoint was to evaluate how RIPC affects different arterial stiffness parameters. Among secondary outcomes was to assess the effect of RIPC on different end-organ damage markers. Our study design and experimental methodology were developed primarily with these goals in mind. However, as another secondary outcome, we investigated the effect of RIPC on the metabolome and assessed correlations between metabolomic and heart and kidney injury markers to gain better insights into RIPC's potential protective mechanisms. This study is valuable since it combines actual clinical data and novel metabolic features. Also, the randomized controlled trial is methodologically the best format in clinical research to evaluate the effects of RIPC, and all the aspects of the study protocol are thoroughly described in the manuscript. In conclusions we have also underlined the necessity of future research for a definite understanding of RIPC molecular mechanisms.
We believe that based on our study design, we pointed out that these results are descriptive and hypothesis-generating rather than confirmatory. We present our novel experimental findings, combine and compare them with current knowledge and hopefully can point towards further research directions.